# Dietary Habits Modify the Association of Physical Exercise with Cognitive Impairment in Community-Dwelling Older Adults

**DOI:** 10.3390/jcm11175122

**Published:** 2022-08-30

**Authors:** Kai Wei, Junjie Yang, Shaohui Lin, Yi Mei, Na An, Xinyi Cao, Lijuan Jiang, Chi Liu, Chunbo Li

**Affiliations:** 1Department of Traditional Chinese Medicine, Shanghai Mental Health Center, Shanghai Jiao Tong University School of Medicine, Shanghai 200030, China; 2Shanghai Institute of Traditional Chinese Medicine for Mental Health, Shanghai 201108, China; 3Shanghai Key Laboratory of Psychotic Disorders, Shanghai Mental Health Center, Shanghai Jiao Tong University School of Medicine, Shanghai 200030, China; 4Department of Geriatrics, Shanghai Ninth People’s Hospital, Shanghai Jiao Tong University School of Medicine, Shanghai 200011, China; 5Clinical Neurocognitive Research Center, Shanghai Mental Health Center, Shanghai Jiao Tong University School of Medicine, Shanghai 200030, China; 6Department of Geriatrics Center & National Clinical Research Center for Aging and Medicine, Jing’an District Central Hospital of Shanghai, Fudan University, Shanghai 200040, China; 7Institute of Psychology and Behavioral Science, Shanghai Jiao Tong University, Shanghai 200030, China; 8CAS Center for Excellence in Brain Science and Intelligence Technology (CEBSIT), Chinese Academy of Sciences, Shanghai 200031, China; 9Brain Science and Technology Research Center, Shanghai Jiao Tong University, Shanghai 200030, China

**Keywords:** dietary habits, physical exercise, cognitive impairment, cohort study, effect modification

## Abstract

Background: Previous studies have confirmed that both healthy diets and physical exercise have preventive effects with respect to cognitive decline with aging. The aim of this study was to investigate whether the associations of physical exercise with cognitive impairment differ in community-dwelling older adults with different dietary habits. Methods: In the 2008/2009 wave of the Chinese Longitudinal Healthy Longevity Survey, 14,966 community-dwelling older adults (≥65 years) were included for analyses. Dietary habits (including daily intake of fruits, vegetables, tea, meat, fish, eggs, food made from beans, salt-preserved vegetables, sugar, garlic, milk products, nut products, mushroom or algae, vitamins and medicinal plants) and physical exercise were assessed. Cognitive impairment was evaluated using the Chinese version of the MMSE in the 2008/2009 and 2011/2012 waves. The effect modifications of physical exercise on cognitive impairment by dietary habits were estimated using logistic regression models. Results: Older adults who practiced physical exercise exhibited a trend of decreased probability of cognitive impairment at baseline and follow-up (OR = 0.92, 95% CI = 0.80–1.06, *p* = 0.273; OR = 0.83, 95% CI = 0.65–1.05, *p* = 0.123, respectively) compared with those who did not practice physical exercise. When stratified by dietary habits, physical exercise had a protective effect with respect to prevalent cognitive impairment in older adults who ate fruits (OR = 0.74, 95% CI = 0.58–0.94, *p* = 0.016), ate food made from beans (OR = 0.76, 95% CI = 0.62–0.93, *p* = 0.007), did not eat sugar (OR = 0.81, 95% CI = 0.68–0.98, *p* = 0.028) and ate milk products (OR = 0.75, 95% CI = 0.57–0.97, *p* = 0.030); in the longitudinal analyses, physical exercise had a protective effect with respect to incident cognitive impairment in older adults who ate fruits (OR = 0.63, 95% CI = 0.41–0.98, *p* = 0.040) and milk products (OR = 0.57, 95% CI = 0.35–0.94, *p* = 0.027). Fruits, food made from beans and milk products modified the associations of physical exercise with prevalent cognitive impairment (*p* values for interaction = 0.008, 0.005 and 0.082, respectively). Conclusions: The associations of physical exercise with cognitive impairment could be modified by certain dietary habits. Physical exercise was not found to be significantly protective with respect to cognitive impairment in older adults unless they had specific dietary habits. Thus, dietary habits should be emphasized when investigating the beneficial effects of physical exercise on cognitive function in community-dwelling older adults.

## 1. Introduction

Cognitive impairment poses a major global public health challenge due to its increasing prevalence in line with population aging [1]. Mild cognitive impairment (MCI) is described as an intermediate stage between the expected cognitive decline of normal aging and that of dementia [2], with up to 50% converting to dementia within 5 years, i.e., 10–15% per year, which is much higher than the 1–3% conversion rate in cognitively normal older people [3]. In China, MCI accounts for approximately 20% in older people > 60 years, indicating that the risk of dementia in Chinese older adults is relatively high [4]. Epidemiologic studies have found that impaired cognition in MCI older adults may spontaneously reverse to normal or near-normal cognition during follow-up, although the risk of converting to dementia is still high [5]. The transition from MCI to dementia has become a significant global public health threat and one of the costliest burdens on health services [6]. Therefore, it is critical to identify effective interventions that can prevent MCI in older adults, as well as delay its progression to dementia.

The aging brain retains considerable functional plasticity, which is very much dependent on the lifestyle of individuals [7]. Lifestyle factors are powerful instruments to promote healthy and successful aging of the brain and delay the appearance of age-related cognitive deficits in older people [8]. It is becoming apparent that successful brain aging is possible if people maintain certain healthy lifestyle habits throughout their lives [7,8]. Diet has been recognized as a modifiable lifestyle factor that may help protect the brain and improve global cognition in later life on a daily basis [9]. Studies focusing on a range of single nutrients (such as n-3 polyunsaturated fatty acid (PUFA)) and healthy dietary patterns (such as dietary approaches to stop hypertension (DASH) and a Mediterranean diet) have been performed and confirmed their protective roles in cognitive decline during aging [9,10]. Although investigations into single nutrients and fixed dietary patterns have importance from the mechanistic point of view, in everyday situations, foods may be more likely to be consumed in varied and complex combinations by individuals, especially in China, where people living in different regions have considerably different dietary cultures and lifestyles. The role of physical exercise, another daily lifestyle factor, in reducing cognitive decline associated with aging was demonstrated in human studies more than 20 years ago [11]. Physical exercise has been shown to promote not only healthy aging of the brain but also delay the onset and slow down the progression of neurodegenerative diseases, including Alzheimer’s disease, the most prominent type of dementia in older adults [12,13]. Adults who exercise throughout life, especially during middle age, exhibit better cognitive performance during later life than adults who do not exercise, preserving their cognitive functions for longer [14].

It has been found that physical exercise has healing effects on aging-related cognitive decline through similar mechanisms as a healthy diet [15]. Interestingly, physical exercise has also been shown to interact with both dietary interventions, boosting the positive effects of a healthy diet and attenuating the unhealthy effects of a poor diet [16]. In turn, some dietary factors, such as curcumin, may contribute to the effects of physical exercise in improving cognitive abilities [17]. To date, few studies have investigated whether the impacts of physical exercise on cognitive impairment differ in terms of daily dietary habits in Chinese community-dwelling older adults, and it remains unclear which diet components may modify these associations and the extent to which such modifications occur. Therefore, the aim of our study was to assess the associations of physical exercise with prevalent and incident cognitive impairment and to determine the effect modifications of dietary habits (including daily intake of fruits, vegetables, tea, meat, fish, eggs, food made from beans, salt-preserved vegetables, sugar, garlic, milk products, nut products, mushroom or algae, vitamins and medicinal plants) on these associations among community-dwelling older adults aged 65 years or older in China.

## 2. Materials and Methods

### 2.1. Study Design and Participants

The Chinese Longitudinal Healthy Longevity Survey (CLHLS) is an ongoing, prospective cohort study of community-dwelling Chinese older adults [18,19]. It covers the majority of the provinces in China, with the aim of investigating the factors associated with healthy longevity of the Chinese population. Started in 1998, follow-ups have been conducted every 2 to 3 years. To reduce attrition, new participants are continually enrolled as death and loss to follow-up are inevitable. Trained interviewers with a structured questionnaire conduct the survey from door to door. Written informed consent was obtained from all participants and/or their proxy respondents, and the study was approved by the Research Ethics Committee of Peking University (IRB00001052-13074). All methods were performed in accordance with the relevant guidelines and regulations.

In the 2008/2009 wave (baseline), 16,954 older adults were initially interviewed, which represents most participants in a single wave of this project to date. We excluded 378 participants younger than 65 years, 308 participants living in an institution (as they were much older (93.1 ± 9.1 years), and institution living differs from community dwelling), 914 participants with diagnosed stroke and 388 with dementia, ultimately including 14,966 community-dwelling older adults for analyses, among whom 51.3% (7675/14,966) survived, 33.3% (4988/14,966) died and 15.4% (2303/14,966) were lost during the 3-year follow-up (the 2011/2012 wave).

### 2.2. Measurements

We used the data of dietary habits and physical exercise at baseline and assessed the associations of physical exercise with cognitive impairment in the total sample and stratified by dietary habits at baseline and 3-year follow-up.

#### 2.2.1. Assessment of Dietary Habits

Older adults’ dietary habits include current daily intake of fruits, vegetables, tea, meat, fish, eggs, food made from beans, salt-preserved vegetables, sugar, garlic, milk products, nut products, mushroom or algae, vitamins and medicinal plants. The frequency of eating fruits and vegetables included ‘every day/almost every day’, ‘quite often’, ‘occasionally’ and ‘rarely or never’. For the purpose of statistical analysis, we recoded the responses into a dichotomous variable: ‘every day/almost every day’ and ‘quite often’ were defined as “Eat”, ‘occasionally’ and ‘rarely or never’ were defined as “Don’t Eat”. The frequency of eating other dietary components included ‘almost every day’, ‘not every day, but at least once per week’, ‘not every week, but at least once per month’, ‘not every month, but occasionally’ and ‘rarely or never’. For the purpose of statistical analysis, we also recoded the responses into a dichotomous variable: the former two as “Eat” and others as “Don’t Eat”.

#### 2.2.2. Assessment of Physical Exercise

Physical exercise was assessed via the question, “Do you take physical exercises regularly at present?” (referring to purposive fitness activities, such as walking, playing ball, running, Qigong, etc.), with answers “yes” or “no”.

#### 2.2.3. Cognitive Impairment

The CLHLS used the Chinese version of the Mini-mental State Examination (MMSE), the validity and reliability of which have been verified [18,19], as a measure of cognitive function in each wave. The total scores range from 0 to 30, with higher scores representing better cognitive function. We used education-adjusted criteria to define “cognitive impairment”: for participants without formal education, an MMSE score ≤ 17 was defined as cognitive impairment; for those with 1–6 years of education, an MMSE score ≤ 20 was defined as cognitive impairment; for those with more than 6 years of education, an MMSE score ≤ 24 was defined as cognitive impairment [20,21].

#### 2.2.4. Covariates

Measures of sociodemographic characteristics at baseline included age, gender, race (Han Chinese or minority), marital status (married or single/separated/divorced/widowed (SDW)), residence (rural or city/town), occupation (non-professional or professional), education (<1 year or ≥1 year), BMI, smoking, alcohol drinking and socioeconomic status (including sufficient financial support, economic independence, adequate medical service and public medical payment). Living preference was assessed via the question, “What kind of living arrangement do you like best?”, with answers “preferring living alone” and “preferring living with children”. Living arrangement was assessed using the question, “Who do you live with?”, with responses ‘living with family (including housemaid)’ and ‘living alone’. Loneliness was assessed via the question, “Do you feel lonely or isolated?”, with answers “lonely” and “not lonely”. Social/leisure activity score was calculated as in a previous study [20], with a higher score representing a higher frequency of social and leisure activities. Self-reported health was assessed via the question, “How do you rate your health status?”, with answers ‘bad’ and ‘very bad’ defined as “poor self-reported health”. Interviewer-rated health was assessed by interviewers, with ‘moderately ill’ and ‘very ill’ defined as “poor interviewer-rated health”. Comorbidity was assessed as a function of whether the respondent was suffering from 24 common chronic diseases listed in the questionnaire (e.g., heart diseases, stroke, diabetes, hypertension, cancers, cataracts and Parkinson’s disease). Serious illness in the past 2 years was defined as “illness that causes hospitalization or being bedridden all the year around”. Hearing and visual ability were also assessed. The Katz Basic Activities of Daily Living (ADL) Scale and the Lawton Instrumental Activities of Daily Living (IADL) Scale were used to assess participants’ physical function. Having difficulty in performing any one or more of the ADL or IADL tasks was defined as functional limitation.

### 2.3. Statistical Analysis

Categorical variables were presented as numbers (percentages), and continuous variables were presented as means (SD). Differences in the distribution of categorical variables among groups were tested by χ^2^ test. For continuous variables, the F test or Kruskal–Wallis test was used for comparison between groups. Logistic regression models were performed to assess the associations of physical exercise with cognitive impairment in the total sample and stratified by dietary habits to calculate the corresponding odds ratios (ORs) and 95% confidence intervals (CIs). To test whether dietary habits were effect modifiers, interaction terms between dietary habits and physical exercise with respect to prevalent cognitive impairment were assessed in logistic regression models adjusted for baseline values of age, gender, race, marital status, residence, occupation, BMI, smoking, alcohol drinking, socioeconomic status, other dietary habits, loneliness, living arrangements, living preference, social/leisure activity score, poor self-rated health, poor interviewer-rated health, comorbidities (≥2), serious illness in the past 2 years, hearing problems, visual impairment and functional limitation. OR estimates of prevalent cognitive impairment were adjusted for the same set of confounders cited above. As many variables changed from 2008/2009 to 2011/2012, interaction terms and OR estimates for incident cognitive impairment were adjusted for age, gender, race, occupation, other dietary habits, stroke at follow-up and changes in other variables from 2008/2009 to 2011/2012. The multicollinearity of the covariates adjusted in the above regression models was assessed by calculating their variance inflation factor (VIF) values (<10 indicating no collinearity). The acceptable level of significance was set as two-sided *p* < 0.05. Stata version 14.0 (StataCorp LP, College Station, TX, USA) was used for data analysis.

## 3. Results

### 3.1. Baseline Characteristics by Physical Exercise and Cognitive Impairment

As shown in Table 1, compared with older adults who did not practice physical exercise, those who practiced physical exercise were younger and less likely to be female, SDW, living in a rural area, to have poor self-/interviewer-rated health, hearing problems, visual impairment and functional limitation and more likely to be engaged in a professional occupation with ≥1 year education, be a current smoker, drink alcohol, not be lonely, prefer living alone and have higher socioeconomic status; furthermore, more older adults who practiced physical exercise, compared with those who did not, ate fruits, vegetables, tea, meat, fish, eggs, food made from beans, salt-preserved vegetables, garlic, nut products, mushroom or algae, vitamins and medicinal plants and had ≥2 comorbidities; they also had higher BMI and social/leisure activity scores. However, the baseline characteristics of cognitively impaired older adults showed inverse trends. Furthermore, older adults who practiced physical exercise and were cognitively impaired were more likely to be Han Chinese, eat sugar and milk products and have suffered a serious illness in the past 2 years.

### 3.2. Effect Modifications of Physical Exercise on Cognitive Impairment by Dietary Habits

As shown in Table 2, in cross-sectional analyses, adjusted for confounders, older adults who practiced physical exercise exhibited a tendency of decreased probability of cognitive impairment at baseline (OR = 0.92, 95% CI = 0.80–1.06, *p* = 0.273). When stratified by dietary habits, compared with older adults who did not practice physical exercise, physical exercise was significantly associated with decreased risk of prevalent cognitive impairment in those who ate fruits and food made from beans, did not eat sugar and ate milk products (OR = 0.74, 95% CI = 0.58–0.94, *p* = 0.016; OR = 0.76, 95% CI = 0.62–0.93, *p* = 0.007; OR = 0.81, 95% CI = 0.68–0.98, *p* = 0.028; OR = 0.75, 95% CI = 0.57–0.97, *p* = 0.030, respectively). Fruits and food made from beans significantly modified the associations of physical exercise with prevalent cognitive impairment (*p* values for interaction = 0.008 and 0.005, respectively). Milk products had the tendency to modify the association of physical exercise with prevalent cognitive impairment (*p* value for interaction = 0.082).

In the longitudinal analyses, adjusted for confounders including stroke at follow-up and changes in variables from 2008/2009 to 2011/2012, older adults who practiced physical exercise still exhibited a tendency of decreased probability of incident cognitive impairment (OR = 0.83, 95% CI = 0.65–1.05, *p* = 0.123). When stratified by dietary habits, compared with older adults who did not practice physical exercise, physical exercise was still significantly associated with decreased risk of incident cognitive impairment in older adults who ate fruits and milk products (OR = 0.63, 95% CI = 0.41–0.98, *p* = 0.040; OR = 0.57, 95% CI = 0.35–0.94, *p* = 0.027, respectively). Dietary habits did not significantly modify the association of physical exercise with incident cognitive impairment (*p* values for interaction = 0.124–0.995).

Age and gender also affected the associations of physical exercise with cognitive impairment in the total sample and stratified by dietary habits. As shown in Appendix A, in the total sample, physical exercise was protective against incident cognitive impairment only in older adults < 80 years (OR = 0.47, 95% CI = 0.28–0.81, *p* = 0.007). When stratified by dietary habits, the associations of physical exercise with prevalent cognitive impairment remained significant in older adults ≥ 80 years who ate fruits (OR = 0.76, 95% CI = 0.59–0.98, *p* = 0.031) and food made from beans (OR = 0.74, 95% CI = 0.60–0.92, *p* = 0.005), did not eat sugar (OR = 0.79, 95% CI = 0.65–0.97, *p* = 0.021) and ate milk products (OR = 0.76, 95% CI = 0.58–1.00, *p* = 0.052); in older men who ate food made from beans (OR = 0.72, 95% CI = 0.53–0.98, *p* = 0.034); and in older women who ate fruits (OR = 0.66, 95% CI = 0.48–0.93, *p* = 0.016), did not eat sugar (OR = 0.72, 95% CI = 0.56–0.93, *p* = 0.012) and ate milk products (OR = 0.69, 95% CI = 0.48–1.01, *p* = 0.054). Furthermore, the associations of physical exercise with incident cognitive impairment remained significant in older adults < 80 years who ate fruits (OR = 0.14, 95% CI = 0.04–0.51, *p* = 0.003) and did not eat sugar (OR = 0.31, 95% CI = 0.14–0.68, *p* = 0.004) and in older men who did not eat sugar (OR = 0.54, 95% CI = 0.31–0.92, *p* = 0.024) and ate milk products (OR = 0.27, 95% CI = 0.16–0.63, *p* = 0.002).

## 4. Discussion

Community-dwelling older adults who practiced physical exercise exhibited a trend of decreased probability of cognitive impairment at both baseline and follow-up compared with those who did not practice physical exercise. In older adults < 80 years, physical exercise was significantly associated with decreased incidence of cognitive impairment at follow-up, and such a trend was also found in older men. These results are consistent with those of a previous study [14], indicating that older adults who practice physical exercise at a relatively younger age exhibit improved cognitive performance over time. Older men might also benefit more from physical exercise than older women in terms of cognitive function during later life. When stratified by dietary habits and adjusted for covariates including daily intake of other diet components, physical exercise was found to be significantly protective against prevalent cognitive impairment in older adults who ate fruits and food made from beans, did not eat sugar and ate milk products and remained protective with respect to incident cognitive impairment in older adults who ate fruits and milk products. Furthermore, fruits, food made from beans and milk products were found to modify the associations of physical exercise with prevalent cognitive impairment.

Emphasizing the beneficial effects of physical exercise as a lifestyle, a number of studies have demonstrated that physical exercise has the capacity to enhance learning and memory under a variety of conditions, from counteracting the cognitive decline that comes with age [15,22] to facilitating functional recovery after brain injury and disease [23]. The effectiveness of physical exercise on cognition may be significantly influenced by frequency, intensity, time, type and period [24]. A meta-analysis involving 4401 older adults aged around 70 years suggested that physical exercise with a frequency ≥3 times/week (vs. 1–2 times/week), vigorous intensity (vs. moderate-intensity), 30–60 min/session (vs. ≤30 min/session), the type of muscle-strengthening exercise (vs. mind-body activity and aerobic activity) and a short period of 6–12 weeks (vs. ≥24 weeks) has a positive effect in terms of cognitive function [24]. However, the authors of some studies concluded that all types and all levels of physical exercise played a significant protective role against cognitive decline [24,25], whereas other studies reported inconsistent findings. A dose–response relationship between physical activity and cognitive function was found in U.S. older adults (mean age: 69 years), with higher physical activity (both daily accumulated and peak effort) associated with better cognitive function [26]. Kim et al. found that a higher frequency of physical exercise could prevent MCI from progressing to dementia [27]. In a controlled trial, aerobic exercise, such as brisk walking for 40 min three times/week, reduced brain atrophy and improved memory and other cognitive functions [28]. Northey et al. found that medium-duration (45 min–60 min) and moderate- or vigorous-intensity exercise interventions showed a significant effect on cognitive function in people aged >50 years [29]. Sanders et al. reported that exercise with a shorter duration (<30 min) and higher session frequency (≥3 sessions/week) predicted more significant effects on cognitive function in older adults with cognitive impairment [30]. One study even showed that moderate levels of physical activity were associated with a relative reduction in health benefits [31].

Diet has been recognized as another modifiable lifestyle factor that may help to improve cognition in later life on a daily basis [9]. A healthy diet is significantly associated with cognitive decline and progression to dementia, showing a protective role against the harmful effects of neuroinflammation and oxidative stress [32]. It is thought that antioxidants in foods such as fruits and vegetables help to reduce oxidative stress levels in the brain, and n-3 PUFAs in foods such as fish are additionally linked to reduced neuro-inflammation [33]. However, a poor diet, such as a diet rich in saturated fat and sugar, can have the opposite effect, elevating levels of oxidative stress and decreasing synaptic plasticity in the hippocampus, impairing learning and memory and increasing the rate of decline in cognitive ability with age [17,34]. The DASH and Mediterranean dietary patterns are both characterized by a moderate-to-high intake of fruits, vegetables, whole grains, poultry, fish and nuts, and reduced intake of red meat. Intake of low-fat dairy products and reduced intake of sweets, sugar-sweetened beverages and sodium are also included in the DASH diet [35]. The Mediterranean diet also includes a high intake of legumes and a moderate intake of red wine (with meals), with olive oil as the main fat source [36]. Although healthy dietary patterns have been found to play a protective role with respect to cognitive decline during aging [9], one study found that not the DASH dietary pattern but some of its components, including nuts, legumes and whole grains, were significantly correlated with improved performance of older adults on cognition tests [37]. In everyday situations, foods may not usually be consumed by individuals in set patterns but in varied and complex combinations, especially for Chinese older adults living in different regions with differing dietary cultures and lifestyles.

It has been found that physical exercise has healing effects on cognitive decline associated with aging through similar mechanisms as a healthy diet [15], and interacts with different kinds of diets, boosting the positive effects of a healthy diet and counteracting the deleterious effects of a poor diet [16]. In turn, some dietary factors, such as curcumin, may contribute to the effects of physical exercise in improving cognitive abilities [17]. However, a 4-year RCT involving 1401 men and women aged 57–78 years revealed that only the combination of at least moderate-intensity aerobic exercise and a healthy diet may improve cognition in older adults over 4 years, with no effect on cognition observed in association with either of these interventions alone, resistance training alone or the combination of resistance exercise and a healthy diet. The authors of this study concluded that diet did not potentiate the effect of aerobic or resistance exercise on cognition in older adults [38]. In the present study, we obtained inconsistent results: consumption of certain diet components could potentiate the protective effect of physical exercise on cognitive impairment in community-dwelling older adults.

Age and gender also affected the associations of physical exercise with cognitive impairment when stratified by dietary habits. In older adults < 80 years, physical exercise was not significantly associated with prevalent cognitive impairment within each strata of dietary habits; however, with respect to incident cognitive impairment, physical exercise exhibited a protective effect in the strata of many diet components, such as the strata of “don’t eat fruits” and “don’t eat fish”. In older men, physical exercise had a protective effect against prevalent cognitive impairment only in the stratum of “eat food made from beans”; with respect to incident cognitive impairment, however, physical exercise also had a protective effect in the strata of many diet components, including the stratum of “don’t eat fish”. In older adults ≥ 80 years and older women, although the results of cross-sectional analyses were consistent/partially consistent with those for total sample, physical exercise no longer showed any significant association with incident cognitive impairment when stratified by dietary habits. The above results indicate that cognitive benefits from physical exercise could counteract and even exceed the deleterious effects of some poor dietary habits over time in older adults who were relatively younger, as well as in older men. For these populations, emphasis should be placed more on physical exercise to delay cognitive decline.

Our study identified the daily diet components that can modify the associations of physical exercise with cognitive impairment in community-dwelling older adults, with a large total sample size covering the majority of provinces in China, which may guarantee the representativeness of our findings. However, our study is still subject to several limitations. First, physical exercise was assessed via one question without considering different dimensions of exercise, such as frequency, intensity, time, type and period, which could explain why a protective trend for cognitive impairment was only found in older adults who practiced physical exercise. Furthermore, our logistic regression models were rigorous, in that we adjusting all covariates associated with cognitive impairment, which is another possible reason for this insignificant result. Second, dietary habits were only assessed qualitatively but not quantitatively in our study, which might have resulted in insufficient evidence to guide older adults’ daily diets. Third, diet components were separately stratified and investigated in our study but not in a certain combination or as a dietary pattern. Fourth, older adults’ dietary habits and physical exercise changed between baseline and the 3-year follow-up, although we did not consider this change in our longitudinal analyses. Therefore, confusions and an insufficient sample size caused by excessive stratifications could be avoided, although possibly resulting in some bias in our results. In the future research, physical exercise should be investigated on more dimensions; dietary habits should be assessed both qualitatively and quantitatively; a healthy dietary scale or healthy dietary patterns should be formed to assess dietary health in different regions of China based on specific dietary cultures and lifestyles; and the dynamic associations between dietary habits, physical exercise and cognitive impairment should be explored to promote successful aging with respect to daily modifiable lifestyle factors.

## 5. Conclusions

The associations of physical exercise with cognitive impairment could be modified by certain dietary habits. Physical exercise was not found to be significantly protective with respect to cognitive impairment in older adults unless they had specific dietary habits. Thus, dietary habits should be emphasized when investigating the beneficial effects of physical exercise on cognitive function in community-dwelling older adults.

## Figures and Tables

**Table 1 jcm-11-05122-t001:** Baseline characteristics according to physical exercise and cognitive function.

Characteristic	Total Sample (*n* = 14,966)	No PE	PE	*p*	NC	CI	*p*
		10,861 (72.6)	4105 (27.4)		10,614 (71.1)	4321 (28.9)	
**Sociodemographic**							
Age (years)	87.4 (11.4)	88.8 (11.2)	83.6 (11.1)	<0.001	83.8 (10.8)	95.9 (7.6)	<0.001
Gender (female)	8696 (58.1)	6802 (62.6)	1894 (46.1)	<0.001	5548 (52.3)	3132 (72.5)	<0.001
Race (minority)	930 (6.2)	785 (7.2)	145 (3.5)	<0.001	744 (7.0)	184 (4.3)	<0.001
Marital status (SDW)	10,484 (70.1)	8039 (74.0)	2445 (59.6)	<0.001	6600 (62.2)	3857 (89.3)	<0.001
Residence (rural)	9269 (61.9)	7401 (68.1)	1868 (45.5)	<0.001	6382 (60.1)	2871 (66.4)	<0.001
Occupation (professional)	970 (6.5)	397 (3.7)	573 (14.0)	<0.001	848 (8.0)	120 (2.8)	<0.001
Education (≥1 year)	5401 (36.2)	3254 (30.0)	2147 (52.4)	<0.001	4491 (42.3)	910 (21.1)	<0.001
BMI (kg/m^2^)	20.2 (3.5)	19.9 (3.4)	21.0 (3.7)	<0.001	20.6 (3.5)	19.4 (3.5)	<0.001
Current smoker	2612 (17.5)	1722 (15.9)	890 (21.7)	<0.001	2139 (20.2)	466 (10.8)	<0.001
Current alcohol drinker	2626 (17.6)	1698 (15.3)	928 (22.6)	<0.001	2061 (19.4)	556 (12.9)	<0.001
Living alone	2428 (16.2)	1723 (15.9)	705 (17.2)	0.052	1883 (17.7)	538 (12.5)	<0.001
Prefer living alone	5508 (36.8)	3600 (33.2)	1908 (46.5)	<0.001	4635 (43.7)	862 (20.0)	<0.001
Not lonely	8685 (58.0)	5829 (53.7)	2856 (69.6)	<0.001	7345 (69.2)	1324 (30.6)	<0.001
**Socioeconomic status**							
Sufficient financial support	11,597 (77.5)	8246 (75.9)	3351 (81.6)	<0.001	8453 (79.6)	3121 (72.2)	<0.001
Economic independence	3617 (24.2)	1971 (18.2)	1646 (40.1)	<0.001	3232 (30.5)	377 (8.7)	<0.001
Adequate medical service	13,839 (92.5)	9902 (91.2)	3937 (95.9)	<0.001	9883 (93.1)	3927 (90.9)	<0.001
Public medical payment	1816 (12.1)	930 (8.6)	886 (21.6)	<0.001	1458 (13.7)	352 (8.2)	<0.001
**Dietary habits**							
Fruit consumption	5732 (38.3)	3740 (34.4)	1992 (48.5)	<0.001	4351 (41.0)	1368 (31.7)	<0.001
Vegetable consumption	13,130 (87.7)	9356 (86.2)	3774 (91.9)	<0.001	9638 (90.8)	3465 (80.2)	<0.001
Tea drinking	5424 (36.3)	3695 (34.0)	1729 (42.2)	<0.001	4354 (41.0)	1056 (24.5)	<0.001
Meat consumption	10,003 (66.9)	7035 (64.8)	2968 (72.4)	<0.001	7269 (68.5)	2713 (62.8)	<0.001
Fish consumption	5812 (38.9)	3882 (35.8)	1930 (47.1)	<0.001	4379 (41.3)	1420 (32.9)	<0.001
Egg consumption	10,360 (69.3)	7302 (67.3)	3058 (74.6)	<0.001	7319 (69.0)	3023 (70.0)	0.215
Food made from beans consumption	8215 (54.9)	5728 (52.8)	2487 (60.6)	<0.001	6026 (56.8)	2169 (50.2)	<0.001
Salt-preserved vegetable consumption	5253 (35.1)	3728 (34.3)	1525 (37.2)	0.001	4065 (38.3)	1176 (27.2)	<0.001
Sugar consumption	5459 (36.5)	3882 (35.8)	1577 (38.5)	0.002	3626 (34.2)	1820 (42.2)	<0.001
Garlic consumption	6030 (40.3)	4020 (37.0)	2010 (49.0)	<0.001	4557 (43.0)	1460 (33.8)	<0.001
Milk product consumption	4277 (28.6)	2672 (24.6)	1605 (39.1)	<0.001	2967 (28.0)	1299 (30.1)	0.009
Nut product consumption	1544 (10.3)	897 (8.3)	647 (15.8)	<0.001	1293 (12.2)	249 (5.8)	<0.001
Mushroom or algae consumption	1718 (11.5)	941 (8.7)	777 (19.0)	<0.001	1355 (12.8)	359 (8.3)	<0.001
Vitamin consumption	1427 (9.5)	843 (7.8)	584 (14.2)	<0.001	1048 (9.9)	373 (8.6)	0.019
Medicinal plant consumption	531 (3.6)	271 (2.5)	260 (6.3)	<0.001	426 (4.0)	103 (2.4)	<0.001
**Physical health status**							
Social/leisure activity score (points)	3.20 (3.1)	2.59 (2.7)	4.83 (3.4)	<0.001	3.98 (3.1)	1.30 (2.0)	<0.001
Poor self-reported health	1994 (14.9)	1600 (16.9)	394 (10.1)	<0.001	1391 (13.1)	598 (22.0)	<0.001
Poor interviewer-rated health	2425 (16.2)	2135 (19.7)	290 (7.1)	<0.001	934 (8.8)	1484 (34.3)	<0.001
Comorbidities (≥2)	6318 (42.2)	4345 (40.0)	1973 (48.1)	<0.001	4625 (43.6)	1673 (38.7)	<0.001
Serious illness in the past 2 years	2224 (14.9)	1550 (14.3)	674 (16.4)	0.001	1463 (13.8)	756 (17.5)	0.001
Hearing problems	3874 (25.9)	3297 (30.4)	577 (14.1)	<0.001	924 (8.7)	2940 (68.0)	<0.001
Visual impairment	3074 (20.7)	2585 (24.0)	489 (12.0)	<0.001	1161 (11.0)	1905 (45.2)	<0.001
Functional limitation	7491 (50.1)	6198 (57.1)	1293 (31.5)	<0.001	3754 (35.4)	3716 (86.0)	<0.001

*Note:* Data are presented as *n* (%) or mean (SD). PE, physical exercise; NC, normal cognition; CI, cognitive impairment; SDW, single/separated/divorced/widowed.

**Table 2 jcm-11-05122-t002:** Associations between physical exercise and prevalent and incident cognitive impairment in the total sample and stratified by dietary habits.

	No Physical Exercise		Physical Exercise		OR (95% CI) for the Association between PE and CI within Each Stratum of Diet Component *
	*n* with/without CI	OR (95% CI)	*n* with/without CI	OR (95% CI)	
**Cross-Sectional Analyses ^a^**					
Total Sample ^#^	3696/7142	1.0 (reference)	625/3472	0.92 (0.80–1.06) *p* = 0.273	
Don’t Eat Fruits	2563/4543	1.0 (reference)	389/1720	1.06 (0.89–1.27) *p* = 0.487	1.05 (0.88–1.25) *p* = 0.583
Eat Fruits	1132/2599	1.07 (0.93–1.23) *p* = 0.376	236/1752	0.77 (0.62–0.97) *p* = 0.027	**0.74 (0.58–0.94)** ***p* = 0.016**
Don’t Eat Vegetables	772/728	1.0 (reference)	83/248	0.89 (0.62–1.29) *p* = 0.540	0.77 (0.54–1.12) *p* = 0.174
Eat Vegetables	2923/6414	0.52 (0.44–0.61) *p* < 0.001	542/3224	0.48 (0.39–0.59) *p* < 0.001	0.95 (0.81–1.11) *p* = 0.535
Don’t Drink Tea	2818/4335	1.0 (reference)	443/1922	0.91 (0.77–1.08) *p* = 0.292	0.95 (0.80–1.13) *p* = 0.569
Drink Tea	876/2806	0.88 (0.77–1.01) *p* = 0.076	180/1548	0.84 (0.66–1.06) *p* = 0.134	0.88 (0.69–1.14) *p* = 0.334
Don’t Eat Meat	1423/2390	1.0 (reference)	182/952	0.85 (0.67–1.08) *p* = 0.189	0.86 (0.67–1.10) *p* = 0.220
Eat Meat	2271/4751	0.85 (0.74–0.97) *p* = 0.017	442/2518	0.81 (0.67–0.99) *p* = 0.036	0.94 (0.79–1.12) *p* = 0.489
Don’t Eat Fish	2533/4428	1.0 (reference)	365/1804	0.91 (0.76–1.09) *p* = 0.317	0.95 (0.79–1.15) *p* = 0.607
Eat Fish	1161/2713	1.09 (0.94–1.25) *p* = 0.266	259/1666	1.02 (0.82–1.27) *p* = 0.856	0.87 (0.69–1.09) *p* = 0.217
Don’t Eat Eggs	1135/2410	1.0 (reference)	160/882	0.94 (0.73–1.22) *p* = 0.657	0.88 (0.67–1.14) *p* = 0.333
Eat Eggs	2559/4731	1.08 (0.93–1.24) *p* = 0.312	464/2588	0.98 (0.81–1.19) *p* = 0.870	0.93 (0.78–1.10) *p* = 0.399
Don’t Eat Food Made from Beans	1840/3281	1.0 (reference)	309/1303	1.14 (0.93–1.39) *p* = 0.211	1.11 (0.90–1.36) *p* = 0.328
Eat Food Made from Beans	1854/3859	0.91 (0.80–1.04) *p* = 0.149	315/2167	0.70 (0.57–0.86) *p* = 0.001	**0.76 (0.62–0.93)** ***p* = 0.007**
Don’t Eat Salt-preserved Vegetables	2713/4403	1.0 (reference)	429/2142	0.89 (0.74–1.05) *p* = 0.168	0.88 (0.73–1.04) *p* = 0.139
Eat Salt-preserved Vegetables	981/2737	0.87 (0.75–0.99) *p* = 0.039	195/1328	0.87 (0.69–1.09) *p* = 0.219	1.00 (0.78–1.28) *p* = 0.985
Don’t Eat Sugar	2162/4800	1.0 (reference)	335/2185	0.84 (0.70–1.01) *p* = 0.070	**0.81 (0.68–0.98)** ***p* = 0.028**
Eat Sugar	1532/2341	1.04 (0.91–1.19) *p* = 0.531	288/1285	1.10 (0.89–1.36) *p* = 0.376	1.13 (0.89–1.42) *p* = 0.315
Don’t Eat Garlic	2500/4322	1.0 (reference)	357/1732	0.89 (0.74–1.07) *p* = 0.206	0.89 (0.74–1.07) *p* = 0.204
Eat Garlic	1193/2819	1.06 (0.93–1.22) *p* = 0.378	267/1738	1.04 (0.84–1.28) *p* = 0.739	0.96 (0.76–1.21) *p* = 0.733
Don’t Eat Milk Products	2612/5557	1.0 (reference)	407/2087	1.00 (0.85–1.19) *p* = 0.969	1.01 (0.85–1.20) *p* = 0.908
Eat Milk Products	1082/1584	1.04 (0.89–1.21) *p* = 0.644	217/1383	0.80 (0.63–1.02) *p* = 0.069	**0.75 (0.57–0.97)** ***p* = 0.030**
Don’t Eat Nut Products	3490/6450	1.0 (reference)	579/2867	0.92 (0.80–1.07) *p* = 0.283	0.92 (0.79–1.07) *p* = 0.275
Eat Nut Products	204/691	0.81 (0.62–1.05) *p* = 0.110	45/602	0.75 (0.48–1.18) *p* = 0.217	0.93 (0.53–1.64) *p* = 0.811
Don’t Eat Mushroom or Algae	3415/6481	1.0 (reference)	543/2772	0.94 (0.81–1.09) *p* = 0.422	0.95 (0.82–1.11) *p* = 0.520
Eat Mushroom or Algae	279/659	1.00 (0.79–1.27) *p* = 0.981	80/696	0.80 (0.55–1.17) *p* = 0.247	0.71 (0.45–1.12) *p* = 0.141
Don’t Eat Vitamins	3374/6621	1.0 (reference)	571/2942	0.96 (0.83–1.11) *p* = 0.581	0.95 (0.82–1.10) *p* = 0.506
Eat Vitamins	320/520	1.11 (0.88–1.40) *p* = 0.386	53/528	0.73 (0.48–1.09) *p* = 0.124	0.77 (0.46–1.27) *p* = 0.305
Don’t Eat Medicinal Plants	3607/6958	1.0 (reference)	608/3227	0.93 (0.80–1.07) *p* = 0.301	0.93 (0.80–1.07) *p* = 0.320
Eat Medicinal Plants	87/183	0.56 (0.37–0.84) *p* = 0.005	16/243	0.46 (0.22–0.93) *p* = 0.032	0.51 (0.17–1.54) *p* = 0.232
**Longitudinal Analyses ^b^**					
Total Sample ^#^	834/3268	1.0 (reference)	284/1938	0.83 (0.65–1.05) *p* = 0.123	
Don’t Eat Fruits	551/2040	1.0 (reference)	161/970	0.84 (0.63–1.13) *p* = 0.260	0.94 (0.70–1.28) *p* = 0.716
Eat Fruits	283/1228	0.69 (0.52–0.90) *p* = 0.008	123/968	0.54 (0.37–0.79) *p* = 0.002	**0.63 (0.41–0.98)** ***p* = 0.040**
Don’t Eat Vegetables	79/292	1.0 (reference)	17/123	0.94 (0.39–2.26) *p* = 0.889	-
Eat Vegetables	755/2976	1.70 (1.07–2.69) *p* = 0.023	267/1815	1.39 (0.85–2.28) *p* = 0.193	0.81 (0.63–1.04) *p* = 0.096
Don’t Drink Tea	543/1916	1.0 (reference)	185/1051	0.80 (0.58–1.08) *p* = 0.146	0.82 (0.59–1.13) *p* = 0.228
Drink Tea	291/1351	0.92 (0.71–1.20) *p* = 0.553	99/886	0.80 (0.56–1.15) *p* = 0.234	0.82 (0.55–1.21) *p* = 0.313
Don’t Eat Meat	269/1137	1.0 (reference)	56/536	0.77 (0.49–1.19) *p* = 0.240	0.74 (0.45–1.23) *p* = 0.243
Eat Meat	565/2130	1.24 (0.94–1.64) *p* = 0.129	228/1401	1.06 (0.75–1.49) *p* = 0.759	0.84 (0.63–1.12) *p* = 0.238
Don’t Eat Fish	541/2048	1.0 (reference)	152/1033	0.71 (0.52–0.97) *p* = 0.033	0.74 (0.54–1.03) *p* = 0.074
Eat Fish	293/1219	0.95 (0.72–1.26) *p* = 0.732	132/904	0.98 (0.68–1.40) *p* = 0.896	0.90 (0.60–1.35) *p* = 0.616
Don’t Eat Eggs	274/1124	1.0 (reference)	81/489	0.92 (0.60–1.40) *p* = 0.697	1.17 (0.73–1.86) *p* = 0.509
Eat Eggs	560/2143	0.88 (0.67–1.17) *p* = 0.388	203/1448	0.70 (0.49–0.98) *p* = 0.040	0.76 (0.56–1.02) *p* = 0.070
Don’t Eat Food Made from Beans	376/1581	1.0 (reference)	113/739	0.84 (0.59–1.21) *p* = 0.349	0.94 (0.64–1.38) *p* = 0.770
Eat Food Made from Beans	458/1685	1.33 (1.02–1.72) *p* = 0.033	171/1198	1.08 (0.77–1.52) *p* = 0.658	0.80 (0.58–1.11) *p* = 0.185
Don’t Eat Salt-preserved Vegetables	530/1924	1.0 (reference)	184/1161	0.91 (0.68–1.24) *p* = 0.559	0.85 (0.62–1.17) *p* = 0.320
Eat Salt-preserved Vegetables	304/1342	1.14 (0.87–1.47) *p* = 0.339	100/776	0.80 (0.55–1.16) *p* = 0.230	0.76 (0.50–1.15) *p* = 0.192
Don’t Eat Sugar	532/2240	1.0 (reference)	161/1247	0.73 (0.53–1.00) *p* = 0.050	0.74 (0.53–1.03) *p* = 0.073
Eat Sugar	302/1027	1.22 (0.94–1.58) *p* = 0.136	123/690	1.19 (0.84–1.67) *p* = 0.322	1.01 (0.68–1.49) *p* = 0.978
Don’t Eat Garlic	516/1916	1.0 (reference)	143/931	0.89 (0.64–1.23) *p* = 0.474	0.89 (0.63–1.27) *p* = 0.519
Eat Garlic	318/1351	1.27 (0.98–1.65) *p* = 0.070	141/1006	0.98 (0.70–1.36) *p* = 0.889	0.86 (0.60–1.22) *p* = 0.395
Don’t Eat Milk Products	647/2613	1.0 (reference)	188/1187	0.81 (0.61–1.09) *p* = 0.163	0.84 (0.63–1.13) *p* = 0.251
Eat Milk Products	187/654	0.71 (0.52–0.98) *p* = 0.040	96/750	0.61 (0.41–0.89) *p* = 0.011	**0.57 (0.35–0.94)** ***p* = 0.027**
Don’t Eat Nut Products	763/2877	1.0 (reference)	260/1562	0.85 (0.66–1.10) *p* = 0.218	0.87 (0.67–1.13) *p* = 0.300
Eat Nut Products	71/390	1.14 (0.73–1.78) *p* = 0.562	24/374	0.74 (0.41–1.34) *p* = 0.323	0.48 (0.18–1.27) *p* = 0.141
Don’t Eat Mushroom or Algae	768/2984	1.0 (reference)	246/1531	0.84 (0.65–1.08) *p* = 0.179	0.87 (0.67–1.12) *p* = 0.275
Eat Mushroom or Algae	66/283	0.80 (0.49–1.32) *p* = 0.384	38/404	0.58 (0.33–1.03) *p* = 0.061	1.32 (0.42–4.19) *p* = 0.635
Don’t Eat Vitamins	773/3066	1.0 (reference)	253/1658	0.83 (0.64–1.06) *p* = 0.140	0.83 (0.64–1.08) *p* = 0.161
Eat Vitamins	61/201	0.97 (0.59–1.61) *p* = 0.914	31/279	0.80 (0.45–1.43) *p* = 0.454	0.78 (0.22–2.75) *p* = 0.701
Don’t Eat Medicinal Plants	814/3205	1.0 (reference)	274/1806	0.84 (0.66–1.08) *p* = 0.179	0.84 (0.66–1.08) *p* = 0.179
Eat Medicinal Plants	20/62	1.74 (0.77–3.93) *p* = 0.186	10/131	0.83 (0.35–1.99) *p* = 0.683	-

^a^ Measure of effect modification on multiplicative scale: fruits: OR (95% CI) = 0.68 (0.51–0.91), *p* = 0.008; vegetables: OR (95% CI) = 1.04 (0.70–1.54), *p* = 0.842; tea: OR (95% CI) = 1.04 (0.78–1.39), *p* = 0.798; meat: OR (95% CI) = 1.13 (0.85–1.52), *p* = 0.405; fish: OR (95% CI) = 1.03 (0.78–1.36), *p* = 0.826; eggs: OR (95% CI) = 0.97 (0.72–1.31), *p* = 0.843; food made from beans: OR (95% CI) = 0.67 (0.51–0.89), *p* = 0.005; salt-preserved vegetables: OR (95% CI) = 1.13 (0.85–1.51), *p* = 0.397; Sugar: OR (95% CI) = 1.25 (0.95–1.65), *p* = 0.118; garlic: OR (95% CI) = 1.10 (0.83–1.45), *p* = 0.510; milk products: OR (95% CI) = 0.77 (0.57–1.03), *p* = 0.082; nut products: OR (95% CI) = 1.01 (0.61–1.70), *p* = 0.957; mushroom or algae: OR (95% CI) = 0.85 (0.55–1.31), *p* = 0.458; vitamins: OR (95% CI) = 0.68 (0.43–1.09), *p* = 0.108; medicinal plants: OR (95% CI) = 0.89 (0.39–2.00), *p* = 0.774. Adjusted for age, gender, race, marital status, residence, occupation, BMI, smoking, alcohol drinking, socioeconomic status, other dietary habits, loneliness, living arrangements, living preference, social/leisure activity score, poor self-rated health, poor interviewer-rated health, comorbidities (≥2), serious illness in the past 2 years, hearing problems, visual impairment and functional limitations. Education was not adjusted, as we used education-adjusted criteria to define “cognitive impairment”. ^b^ Measure of effect modification on multiplicative scale: fruits: OR (95% CI) = 0.94 (0.59–1.51), *p* = 0.798; vegetables: OR (95% CI) = 0.87 (0.35–2.15), *p* = 0.763; tea: OR (95% CI) = 1.09 (0.69–1.74), *p* = 0.701; meat: OR (95% CI) = 1.11 (0.66–1.84), *p* = 0.699; fish: OR (95% CI) = 1.43 (0.91–2.27), *p* = 0.124; eggs: OR (95% CI) = 0.86 (0.52–1.41), *p* = 0.540; food made from beans: OR (95% CI) = 0.97 (0.61–1.53), *p* = 0.889; salt-preserved vegetables: OR (95% CI) = 0.77 (0.48–1.22), *p* = 0.266; sugar: OR (95% CI) = 1.34 (0.85–2.11), *p* = 0.213; garlic: OR (95% CI) = 0.87 (0.55–1.36), *p* = 0.539; milk products: OR (95% CI) = 1.04 (0.63–1.72), *p* = 0.869; nut products: OR (95% CI) = 0.77 (0.38–1.56), *p* = 0.463; mushroom or algae: OR (95% CI) = 0.87 (0.43–1.77), *p* = 0.693; vitamins: OR (95% CI) = 1.00 (0.48–2.09), *p* = 0.995; medicinal plants: OR (95% CI) = 0.57 (0.18–1.81), *p* = 0.341. Adjusted for age, gender, race, occupation, other dietary habits, stroke at follow-up, and changes in marital status, residence, BMI, smoking, alcohol drinking, socioeconomic status, loneliness, living arrangement, living preference, social/leisure activity score, self-rated health, interviewer-rated health, comorbidity number, serious illness in the past 2 years, hearing problems, visual impairment and functional limitations from 2008/2009 to 2011/2012. Education was not adjusted, as we used education-adjusted criteria to define “cognitive impairment”. ^#^ All dietary habits were adjusted. * *Note*: OR (95% CI): odds ratio (95% confidence interval); PE: physical exercise; CI: cognitive impairment.

## Data Availability

The datasets used and analyzed during the current study are available from the Peking University Open Research Data Platform (https://opendata.pku.edu.cn/dataverse/CHADS?from=timeline&isappinstalled=0, accessed on 7 March 2020).

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
