# Peer review of "Dietary Habits Modify the Association of Physical Exercise with Cognitive Impairment in Community-Dwelling Older Adults"

_jcm, 2022, doi:10.3390/jcm11175122_

Round 1

Reviewer 1 Report

It is a well-written manuscript that covers an interesting and actual topic. The English language is of sufficient quality.

The major strengths of the manuscript is the large sample of subject included in the analysis.

The major weakness of the paper is that, as they mentioned in the discussion, physical exercise was only assessed via one question without, considering different dimensions of exercise, such as frequency, intensity, time, type and period.

Author Response

We appreciate the reviewer’s comments and agree with the reviewer. This is one of the major limitations in our study. In the future research, physical exercise would be investigated in more dimensions.

Reviewer 2 Report

The authors conducted a cross-sectional and longitudinal analysis of the associations of exercise and nutritional habits on cognitive decline in a representative large Chinese sample.
Major limitation is that the study is not able to proof causality due to it's non-intervantional design and the multiple nutritional factors (N=15) tested rendering associations that may also have been caused by chance. Therefore the conslusions that exercise in combination with some nutritional habits is protective for cognitive decline, as stated by the authors, is not evidenced by their data. Moroever, synergism by exercise and intake of specific nutritional compounds (for example eating fruits or beans in the crosssectional data, and taking fruits or milkproducts) is not substantiated by increase of the odds ratio nor by specific testing for synergistic interactions. This should be added to warrant the conclusions drawn in the discussion.

The discussion  should best berewritten to make it more easily understandable, as in all parts first a discusion of literature is given, instad of first summarizing the results of the current study, and next analyzing these results in the context of the recent studies found in the literature.

As the researchers also note, a major limitation of the study is that multiple nutritional habits (>15) are tested each on its own, though in the introduction the researchers already noted that the Mediterrenean or DASH diet are most powerful and best evidenced so far. Thus, it would have been logic if the nutritional habits would have been grouped and tested in these dietary patterns. This can still be carried out as it strengthens the conclusions, and would limit the pitfall of multiple comparisons across all nutritional components.

Minor points:

In the results sections (first paragraph) it is written: "Cognitively impaired older adults, however, were on the contrary.  which is an unclear message.
In the other parts of the results section readibility is limited by the long enumerations of data, which in fact copy the results presented in the tables. These parts can be limited substantially in length.

Round 2

Reviewer 2 Report

The paper has been improved and now meets the quality limitations of its design.